# Retrieval of Injection Molding Industrial Knowledge Graph Based on Transformer and BERT

Zhe-Wei Zhou [1], Wen-Ren Jong [1,2,3,*], Yu-Hung Ting [1,2,3], Shia-Chung Chen [1,2,3] and Ming-Chien Chiu [4]

1   Department of Mechanical Engineering, Chung Yuan Christian University, Taoyuan City 320, Taiwan;
    g10973013@cycu.edu.tw (Z.-W.Z.); august@cycu.edu.tw (Y.-H.T.); shiachun@cycu.edu.tw (S.-C.C.)
2   R&D Center for Smart Manufacturing, Chung Yuan Christian University, Taoyuan City 320, Taiwan
3   R&D Center for Semiconductor Carrier, Chung Yuan Christian University, Taoyuan City 320, Taiwan
4   Gudeng Precision Industrial Co., Ltd., New Taipei City 236, Taiwan; bill@gudeng.com
*   Correspondence: wenren@cycu.edu.tw

**Abstract:** Knowledge graphs play an important role in the field of knowledge management by providing a simple and clear way of expressing complex data relationships. Injection molding is a highly knowledge-intensive technology, and in our previous research, we have used knowledge graphs to manage and express relevant knowledge, gradually establishing an injection molding industrial knowledge graph. However, the current way of retrieving knowledge graphs is still mainly through programming, which results in many difficulties for users without programming backgrounds when it comes to searching a graph. This study will utilize the previously established injection molding industrial knowledge graph and employ a BERT (Bidirectional Encoder Representations from Transformers) fine-tuning model to analyze the semantics of user questions. A knowledge graph will be retrieved through a search engine built on the Transformer Encoder, which can reason based on the structure of the graph to find relevant knowledge that satisfies a user's questions. The experimental results show that both the BERT fine-tuned model and the search engine achieve an excellent performance. This approach can help engineers who do not have a knowledge graph background to retrieve information from the graph by inputting natural language queries, thereby improving the usability of the graph.

**Keywords:** injection molding; knowledge management; BERT; Transformer

## 1. Introduction

Plastic injection molding has multiple advantages and its application scope is gradually expanding, with relevance to the manufacturing processes of daily products and precision components. Jong et al. [1,2], based on accumulated practical knowledge from past mold design processes, established a mold design navigating system using computer-aided design (CAD) secondary development tools and a historical knowledge base. This system can assist personnel in making decisions during the mold design process, accelerate the mold design schedule, and prevent losses caused by an improper design. Khosravani et al. [3] proposed an intelligent injection molding machine defect detection system based on Case-Based Reasoning (CBR). This system can compare current defect situations with historical records, identify similar cases, and eliminate defects based on their solutions, effectively reducing the downtime caused by injection molding machine failures. When encountering molding quality issues during the injection process, Mikos et al. [4] used CBR to find similar historical cases and used the retrieved case solutions as the basis for solving the current problem.

From the above literature, it can be observed that, in the current injection molding industry, problems are often solved based on experience and past cases. Engineers often do not understand the reasons for selecting a particular solution or the associated effects caused by that solution. With the shortening of product life cycles and the increasing

frequency of product design changes, relying on past experience to solve problems may not be sufficient to keep up with current trends. Therefore, building an injection molding knowledge base and providing a suitable and transparent search method for engineers to find relevant knowledge to serve as a basis for their problem solving will be a critical step for continuous improvements in the industry.

With the advancement of technology, the knowledge-based economy era has arrived. Knowledge has become a key factor for enterprises to maintain their competitiveness. Therefore, businesses have gradually realized the importance of knowledge management. In recent years, knowledge graphs have become a core issue in this knowledge management, and many industries have begun to conduct in-depth research on knowledge graphs, achieving many results. Bader et al. [5] established an Industry 4.0 knowledge graph, which recorded the relevant standards, regulations, and reference frameworks of Industry 4.0, helping relevant personnel to understand how to implement Industry 4.0 systems. Shi et al. [6] obtained agricultural Q&A data and popular science data through web crawling to be used as their raw data and established a knowledge graph consisting of three categories: crops, pesticides, and pests. The graph recorded the mutual importance between different crops, the mutual influence between pesticides and pests, and the trend of pest occurrence. It provided good guidance for agricultural production. Chen et al. [7] established a food knowledge graph and proposed a recommendation formula that could consider user dietary preferences and health conditions, recommending the appropriate food information to its users. Chi et al. [8] designed a healthy diet knowledge graph, integrating health and dietary information from the internet and providing a retrieval system. By using this system, users could learn healthy diet knowledge more quickly and comprehensively, and may be more inclined towards balanced diets. Yu et al. [9] established a traditional Chinese medicine knowledge graph and developed corresponding knowledge acquisition techniques. They provided a new analytical method for traditional Chinese medicine knowledge that had broad application prospects in the field of traditional Chinese medicine healthcare. Cui et al. [10] designed a graph-theoretic knowledge model to enhance knowledge management and cited examples from the traditional Chinese medicine research field to illustrate the usability of this model. Their research results showed that the model could promote knowledge learning, sharing, and decision making among researchers. Zhang et al. [11] integrated medical text knowledge with physicians' clinical experiences to create a health knowledge graph, and designed an expandable mechanism for the graph that could add new diseases to the existing knowledge graph. The graph could assist clinical doctors with their diagnostic decision making. Kim [12] constructed a knowledge graph that included all the healthcare facilities in Korea. The graph was interconnected with heterogeneous data sources such as administrative areas, allowing users to choose suitable healthcare facilities based on their requirements. Jiang et al. [13] established a Knowledge Graph of Construction Safety Standard (KGCSS) to guide the development and revision of the Construction Safety Standards (CSS) for building safety. In addition, it also provided a search function. The construction of this KGCSS promoted the analysis, search, and sharing of standard safety knowledge. Gao et al. [14] introduced a construction method for a power system dispatch knowledge graph, which recorded the correlations among various power devices and the impacts of device failures. Through investigating the application of the knowledge graph in the practical field of power system dispatch, it could be proved that the knowledge graph could assist dispatchers with their knowledge management and emergency decision making. Zhang et al. [15] constructed a knowledge graph to express the interrelationships among enterprises, and based on this graph, they searched for related enterprises from the perspective of the target enterprise. They used Graph Neural Networks (GNN) to calculate the credit risk of the target enterprise, which served as the basis for investment decisions. The above literature shows that knowledge graphs have achieved excellent performances in various fields of knowledge management. Unlike traditional knowledge repositories, knowledge graphs can clearly

express the interactive relationships between data and play an important role in knowledge retrieval, question answering, and knowledge recommendations.

Knowledge graphs consist of multiple nodes, and in terms of knowledge analysis, they can observe the paths between two nodes and identify their potential relationships. If the paths between these nodes are understood as sequential data, applying machine learning methods will help to reason and analyze the knowledge graph. Wu et al. [16] used a Transformer model to predict the future trend of influenza transmission based on the past 10 weeks of transmission data. The performance of the Transformer model was compared with traditional sequence models such as LSTM [17] and Seq2Seq [18]. The results showed that the Transformer outperformed LSTM and Seq2Seq, and this model could help public health experts to respond to future changes in an epidemic situation in a timely manner. Grechishnikova [19] used Transformer-based models to input the amino acid sequence data of a target protein and generate molecular structures that could bind to this target protein, serving as the main basis for drug development. This approach significantly reduced the time and cost required for drug development. Ding et al. [20] used Transformer models to predict the remaining lifespan of machines based on vibration signals and compared the performance of this approach with other models, such as CNN, LSTM, and RNN, in the task of machine lifespan prediction. The results showed that the Transformer achieved a better performance than the other models. Zhang et al. [21] trained a Transformer model to analyze the trends of factors such as temperature, light intensity, and carbon dioxide concentration, in order to predict future greenhouse temperatures. The results showed that the predictive performance of this model was better than the existing prediction methods. Hu et al. [22] proposed a model for detecting cardiac arrhythmia based on a CNN and Transformer and verified its performance using the MIT-BIH dataset. The results showed that the model was able to predict the classes of all the heartbeats in the electrocardiogram, and achieved accuracies of 99.12%, 99.49%, and 99.23% in the prediction tasks of eight, four, and two heartbeat classes, respectively. Chen et al. [23] used a CNN-Transformer to consider the historical changes in NO, $NO_2$, $SO_2$, CO, wind speed, wind direction, minimum temperature, and maximum temperature to predict the average ozone concentration. The results showed that the model achieved RMSE values of 7.75 and 16.27 in predicting the ozone concentrations for the next one and three days, respectively. The above studies showed that Transformer models have an excellent understanding of sequential data, further demonstrating the potential of these Transformer models in identifying potential relationships based on node-to-node paths.

Although knowledge graphs can clearly express the relationships between nodes, the current way of searching through knowledge graphs mostly involves writing programming languages. Engineers who lack a background in knowledge graphs may encounter many challenges when searching through them; therefore, providing an intuitive search method can improve the usability of these knowledge graphs. Acikalin et al. [24] fine-tuned a multilingual BERT model and attempted to use it for classifying the positive and negative reviews of Turkish movies and hotels in a review dataset. Their experimental results showed that the model achieved an extremely high accuracy. Mozafari et al. [25] used a fine-tuned BERT model to detect unfriendly language on social media and classify it into themes such as racism, sexism, hate, and attacks. The results showed that the model achieved a good accuracy and could improve the negative effects that current speech monitoring tools bring to platforms and users due to their errors. Zheng et al. [26] used a BERT model with a CRF (Conditional Random Field) for named entity recognition tasks to identify the relevant terms in hazardous chemical accident records and construct a knowledge graph for hazardous chemical management, assisting personnel with controlling these hazardous chemicals. Yu et al. [27] proposed a BERT-BiLSTM-CRF model for entity recognition in electric power safety, which extracted the relevant terms from unstructured data and validated the model's performance. The results showed that the performance of the proposed model was better than that of other models. Harnoune et al. [28] used a BERT model with CRF to extract knowledge from biomedical clinical records and created a medical

knowledge graph. This provided an intuitive way of visualizing medical knowledge and could help medical personnel to quickly find the information they need. Chansai et al. [29] used BERT to extract entities and relationships from dental textbooks and constructed a knowledge graph to help dental students learn the important concepts in the textbooks. The above literature demonstrates that BERT has achieved excellent performances in semantic understanding, entity recognition, and relationship extraction. Therefore, BERT will play an important role in user question understanding.

## 2. Background Knowledge

### 2.1. Knowledge Graph

The concept of a knowledge graph was proposed by Google in 2012 [30]. The basic unit of a knowledge graph is a semantic triple, which consists of entities and the relationships between them. These entities are the nodes in a knowledge graph, which can represent concrete people, things, and objects, as well as abstract concepts. The relationships between these entities are the edges in a knowledge graph that link the nodes together. By assigning directional edges between the nodes, the subject and object of a relationship can be described. Compared to traditional tabular data, knowledge graphs can connect different entities together and provide the ability to understand information from a relational perspective.

### 2.2. Neo4j

Neo4j [31] is one of the most popular graph databases from recent years. Its built-in Cypher language enables the rapid editing of a knowledge graph. Additionally, Neo4j can visualize graphs using D3.js, which helps engineers to clearly view the relationships between the nodes, significantly reducing the maintenance costs of the graph. Therefore, in this study, Neo4j was selected as the database for storing the injection molding industrial knowledge graph.

### 2.3. MongoDB

Although using Neo4j as the database for storing a knowledge graph can reduce the maintenance cost, its retrieval performance is insufficient for the needs of this study. Retrieving the relevant attributes and neighboring node information of any node in a graph through the Neo4j Python Driver takes about 4 s. MongoDB is an NoSQL database that stores data in the JSON format and provides an object-oriented query language for fast data retrieval. Therefore, we use the Neo4j Python Driver to obtain the relevant information of the knowledge graph and organize it. Then, using the pymongo Python package of MongoDB, we store the node properties and node relationships in two collections in MongoDB, as shown in Figure 1. With MongoDB, we only need 0.003 s to complete the search operation, which took Neo4j 4 s.

### 2.4. Transformer Encoder

This study will use the Transformer Encoder [32] to extract node path features. The Transformer Encoder is a neural network architecture based on the Multi Head Self Attention [32] mechanism, specifically designed for reading sequential data and generating features. Its overall structure is shown in Figure 2. When receiving a sequential input, the Encoder first maps it into multidimensional data through the Word Embedding Layer. Due to the interpretative method of Multi Head Self Attention for sequential data, it cannot consider the position of each node in the sequence. Therefore, we need to generate positional encoding to indicate the position of each node in the sequence data, enabling the Multi Head Self Attention to consider the node position when calculating the node features. Additionally, when interpreting the input generated by the Multi Head Self Attention, the Encoder takes the original input into consideration, performs Layer Norm processing, and inputs the processed feature data into the Fully Connected Layer. Then, it uses the mechanism for interpreting the output of the Multi Head Self Attention to process the

output of the Fully Connected Layer, which can effectively interpret sequential data and generate multidimensional features for subsequent applications.

**Node Property Collection**

**_id:** {   "$oid": "637c1dbc6348e7edd1c52b31"  }

**nodeid:** 80

**title:** "Increase cavity pressure"

**layer:** "Phenomenon"

**Node Relation Collection**

**_id:** {   $oid:"637c1dbc6348e7edd1c52c25"  }

**nodeid:** 80

**totalconnect:** [

{      **node:** 79,     **relation:** "cause" },

{      **node:** 42,     **relation:** "cause" },

{      **node:** 90,     **relation:** "may cause" },

{      **node:** 91,     **relation:** "may cause "}

]

**Figure 1.** MongoDB storage format.

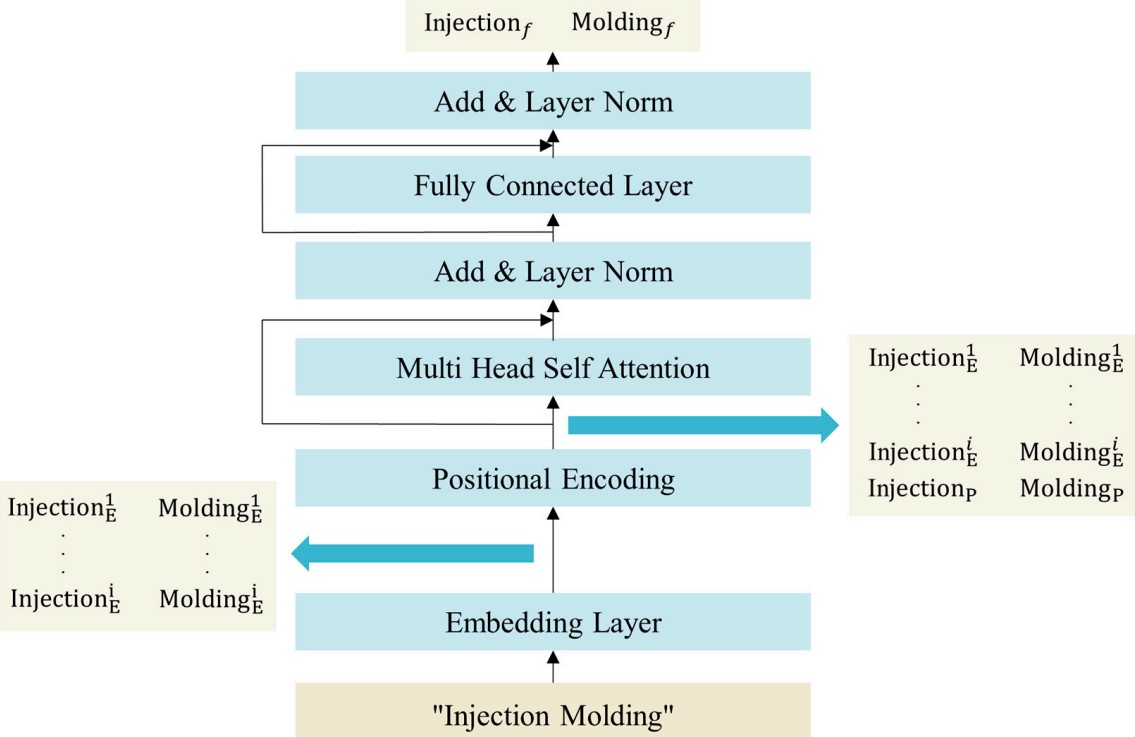

**Figure 2.** Transformer Encoder.

*2.5. BERT*

BERT is a pre-trained natural language model that was proposed by Google in 2018 [33]. It consists of 12 Transformer Encoders and is trained on a large amount of text data, with 1.1 billion weight parameters. BERT is capable of extracting 768-dimensional semantic features from input text data, as shown in Figure 3. Before reading text data, BERT requires preprocessing via a Tokenizer, as shown in Figure 4. This Tokenizer converts the text data into Token Embedding, Segment Embedding, and Position Embedding, which, respectively, record the character encoding, sentence segmentation annotation, and position information of each character in the text data. These three embeddings are used as inputs for BERT to extract the text data features. The purpose of this step is to convert natural language into a form that can be computed by a computer. The biggest difference between BERT and traditional natural language models is that, when extracting features for each character in the text data, BERT considers a character's position and its corresponding context in the text data, which results in an excellent performance in interpreting text data.

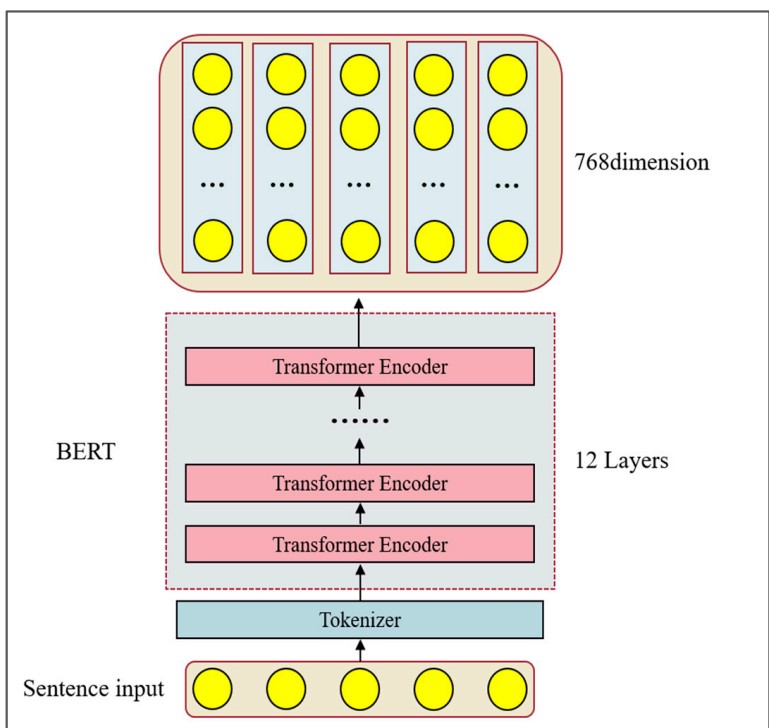

**Figure 3.** BERT.

|  | What causes burrs. How to solve burrs. | | | | | | | | | |
|---|---|---|---|---|---|---|---|---|---|---|
| Token Embedding | $E_{[CLS]}$ | $E_{what}$ | $E_{causes}$ | $E_{burrs}$ | $E_{[SEP]}$ | $E_{How}$ | $E_{to}$ | $E_{solve}$ | $E_{burrs}$ | $E_{[SEP]}$ |
| Segment Embedding | $E_A$ | $E_A$ | $E_A$ | $E_A$ | $E_A$ | $E_B$ | $E_B$ | $E_B$ | $E_B$ | $E_B$ |
| Position Embedding | $E_0$ | $E_1$ | $E_2$ | $E_3$ | $E_4$ | $E_5$ | $E_6$ | $E_7$ | $E_8$ | $E_9$ |

**Figure 4.** Tokenizer encoding example.

## 3. Methodology

### 3.1. Injection Molding Industrial Knowledge Graph

The injection molding industrial knowledge graph [34] adopted in this study uses seven node categories: molding defects, process parameters, phenomena, molds, plastics, product design, and machine. It is accompanied by three types of edges: "causes", "may causes", and "solutions", in order to express injection molding knowledge. The main difference between the two types of edges, "causes" and "may causes", is that the edges of the "causes" type are mainly used to describe the fact that, if A occurs, then B will inevitably occur, such as: if the melt temperature rises, then the melt density will inevitably decrease. The edges of the "may causes" type describe the fact that, under specific circumstances, A will lead to B, such as: when the flow resistance is close to the critical value that the injection pressure can bear, an increase in the melt flow resistance will lead to the injection pressure being unable to overcome the flow resistance.

Figure 5 shows the current injection molding industrial knowledge graph, which consists of 207 nodes and 258 edges. The graph records the causes and solutions of various molding defects in injection molding, as well as the effects of adjusting the process parameters, molds, or product design changes. The injection molding knowledge, which is complex and intertwined, is expressed clearly in the form of a knowledge graph.

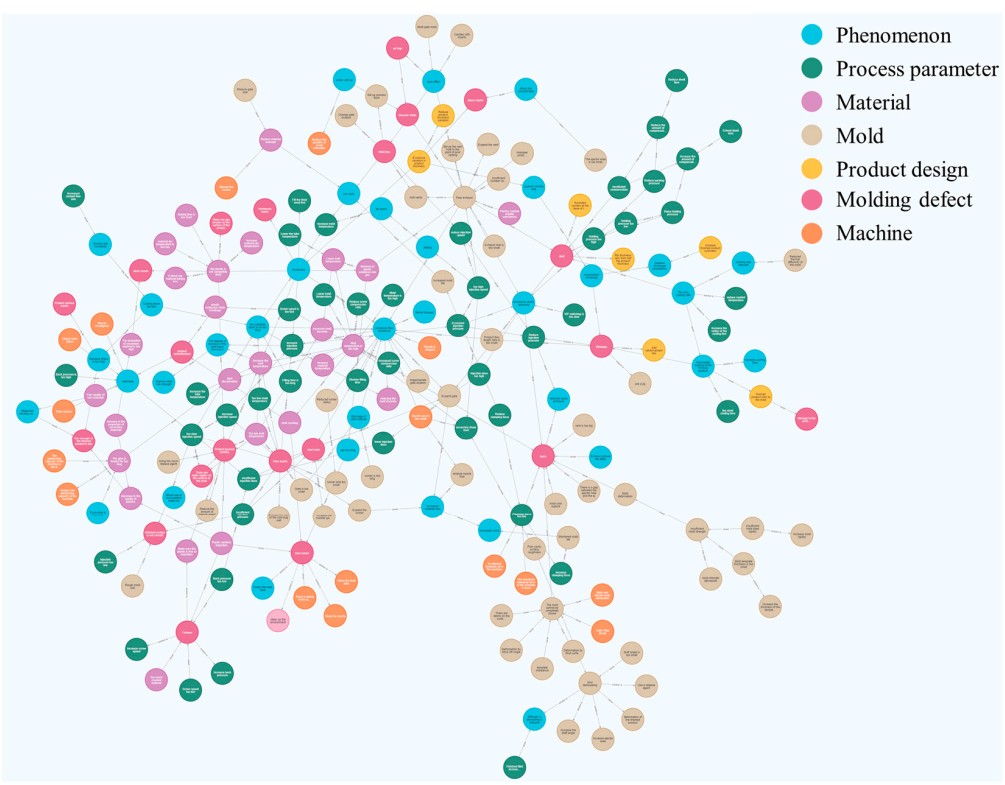

**Figure 5.** Injection molding industrial knowledge graph.

### 3.2. Question Understanding

In our previous study [35], we used self-defined search rules to retrieve information from the knowledge graph and obtained some results. However, the previously developed search mechanism was unable to search the knowledge graph based on the semantics of the user questions, resulting in search results that were flooded with irrelevant knowledge. Therefore, in this study, we analyze the semantics of the user queries using natural language processing technology to make the search results more relevant to a user's question. We break down the understanding of user questions into three tasks: subject

recognition, semantic classification, and entity alignment, which will be introduced in the following order.

### 3.2.1. Subject Recognition

Subject recognition can be explained as identifying the subject of a user's question. For example, if the user inputs the question "Reason for short shot", we hope that the model can successfully identify "short shot" as the subject of the query. Therefore, we use semi-supervised labeling to mark the beginning and end positions of the subject, as shown in Figure 6, and use this as the training data label.

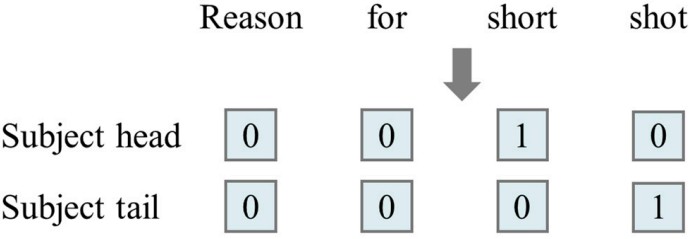

**Figure 6.** Semi-supervised labeling.

Figure 7 shows the workflow of the subject recognition model. A user's question is encoded by the Tokenizer and then input to BERT for the text feature extraction. The extracted text features are processed by the Normalization Layer and then input to the fully connected layer with a Sigmoid activation function, which produces a 2-dimensional array to predict the head and tail positions of the subject.

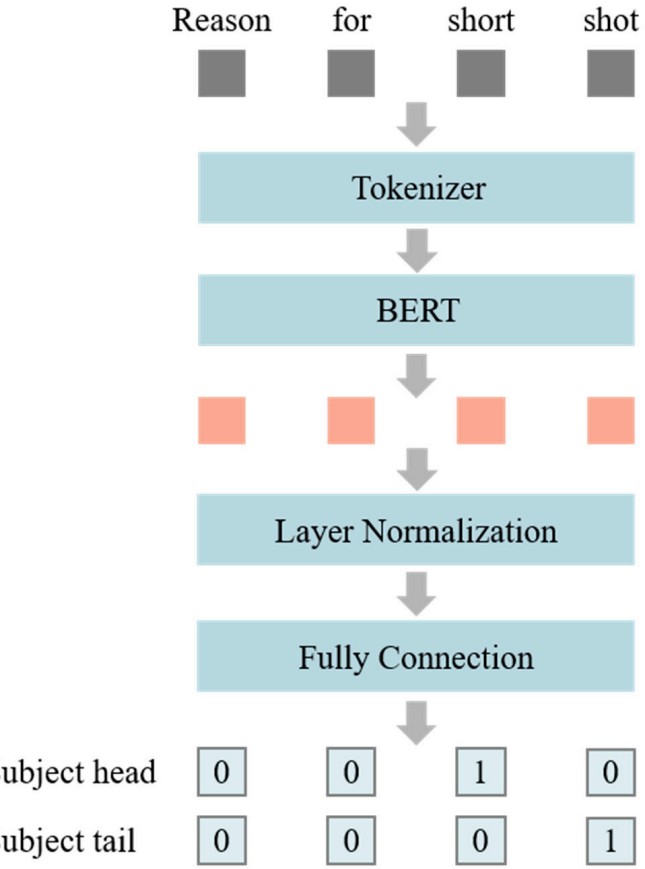

**Figure 7.** Architecture of subject recognition model.

This study prepares 1200 training data, which are established by collecting the common vocabulary in injection molding and combining them with common question grammar. We use Binary Cross-entropy as the loss function for the model and the Adam optimizer to train the model, with a learning rate of 0.00001 and a batch size of 32. After 100 rounds of training, the model achieves an accuracy of 0.953 on 150 test data. The dataset we prepared encompasses the common vocabulary and question syntax in injection molding, such as "How do I deal with short shots?" and "I want to know information about short shots." These common user queries have been included in the training dataset. Therefore, the performance obtained by the model for the test data demonstrates its capability for practical applications. However, in the future, when encountering more obscure terms or complex grammar structures, the model may exhibit recognition errors. To enhance the model's performance, it will be necessary to increase the volume of the training data.

### 3.2.2. Semantic Classification

After successfully identifying the subject from a user's question, it is necessary to filter the subject-related knowledge based on the semantic meaning of the question. For example, for question like "What are the solutions for short shots?", instead of finding all the related knowledge about "short shots", if we can find the knowledge that can solve the problem of "short shots" from the search results, it will make these search results more relevant to the user's needs.

This study categorizes the semantic meaning of user questions into four types: impact, reason, solution, and all relevant knowledge. We train a semantic classification model using the same dataset as the subject recognition model, consisting of 1200 training examples and 150 validation examples. The semantic labels for the training data are represented using a one-dimensional one-hot vector of length 3, and the overall labeling scheme is shown in Table 1.

**Table 1.** Semantic classification label.

| Class | Label |
|---|---|
| Impact | [1,0,0] |
| Solution | [0,1,0] |
| Reason | [0,0,1] |
| All | [0,0,0] |

We classify user queries that involve the impact of a subject on a user's question semantics as "Impact." User queries seeking relevant knowledge to improve this subject are classified as "Solution". Additionally, queries inquiring about the reasons behind the subject's existence are categorized as "Reason". Finally, queries seeking all relevant knowledge about the subject are classified as "All".

Figure 8 shows the architecture of the semantic classification model. We use the user's question and the position of its subject as inputs to the model. The question is encoded by the Tokenizer and then input into BERT to extract the text features. After obtaining these text features, the model extracts the corresponding segment of the text features based on the position of the Subject, which is used as an input to the conditional layer normalization function, in order to process the text features generated by BERT. Conditional Layer Normalization can have different understandings of text data features based on the input of the conditional function, in order to cope with future user questions containing multiple subjects, and to judge their corresponding semantics based on different subjects. The features processed by Conditional Layer Normalization are input to the fully connected layer with Sigmoid as its activation function, outputting a one-dimensional array to represent the semantic classification results.

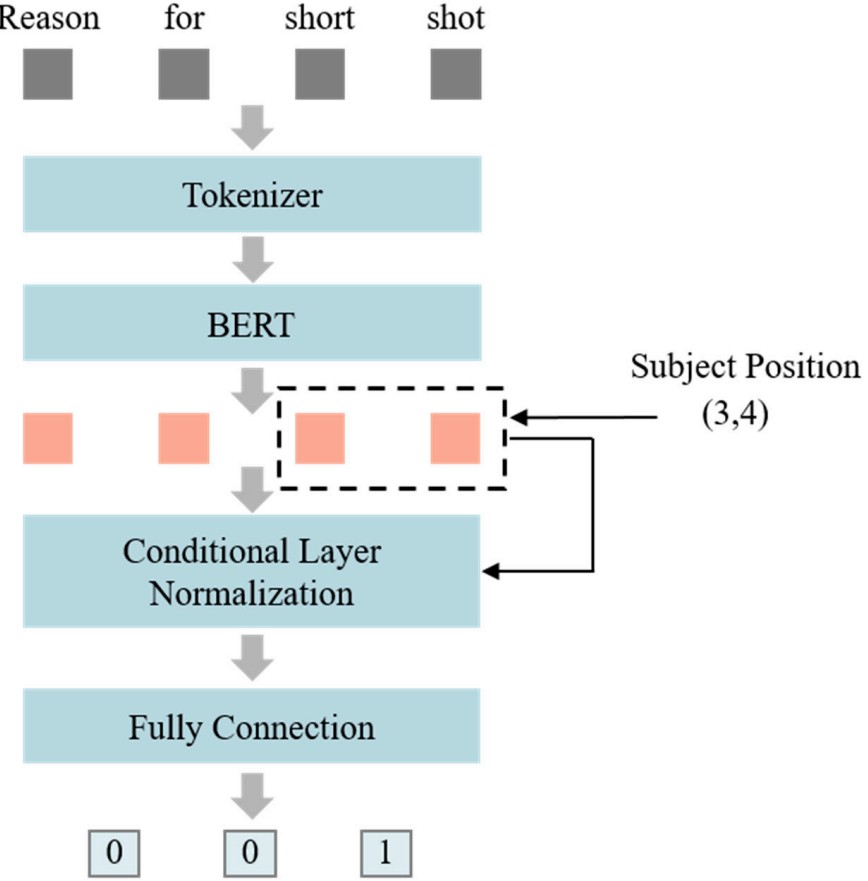

**Figure 8.** Architecture of semantic classification model.

We use Binary Cross-entropy as the loss function and Adam as the optimizer for the model. With a learning rate of 0.00001 and a batch size of 32, the model is trained for 100 epochs. The model achieves a 100% accuracy on 150 test data. As mentioned earlier, the dataset prepared for this study includes the common vocabulary and question grammar in injection molding. Therefore, we can demonstrate the model's ability to accurately discern the semantics of common user questions through its performance on the validation dataset. While the model showed an excellent performance in our testing, there is still the possibility of semantic misjudgment when facing more complex grammar in user queries in the future. Additionally, the current training dataset only includes user questions with a single subject, which means the model is not yet equipped to handle tasks involving user questions with multiple subjects. In our previous study [34], we verified that Conditional Layer Normalization can assist natural language models with understanding text data based on different subjects, thereby increasing our confidence in the semantic classification model's ability to successfully handle user question semantics involving multiple subjects as the training data increase.

### 3.2.3. Entity Alignment

In the field of injection molding, there are often situations where the same concept is expressed in multiple ways, such as "excessive flow resistance" and "flow resistance is too high" both expressing the same concept. When users search a knowledge graph, if the system cannot find the corresponding node in the graph according to the concept expressed by the subject, the retrieval performance will be reduced.

In the past, Levenshtein distance was often used as a basis for determining whether two sentences had similar meanings. Levenshtein distance is calculated as the minimum number of operations needed to transform one sentence into another through insertion,

deletion, or substitution. However, there are many cases where Levenshtein distance cannot be used to determine whether two sentences have similar meanings. For example, the entity pairs "decrease injection speed" and "increase injection speed" have a very low Levenshtein distance but have different meanings, while the entity pairs "The product is unable to achieve complete filling" and "short shot" have a very high Levenshtein distance but have the same meaning. These examples demonstrate that it is difficult to rely on Levenshtein distance between entity pairs to accomplish entity alignment tasks.

Therefore, in this study, we utilize the SBERT (Sentence-BERT) model, as employed in our previous research [34], to identify the nodes in the knowledge graph that have similar concepts to the subject of the user's question. Figure 9 shows the architecture of the SBERT model. When determining the similarity of concepts between the subject and nodes, the model extracts text features individually using BERT, processes the features through Mean Pooling, projects them onto a vector space, and calculates their cosine similarity as a basis for judging whether they have the same concept. In the vector space, entities with similar concepts have higher cosine similarity values, and vice versa.

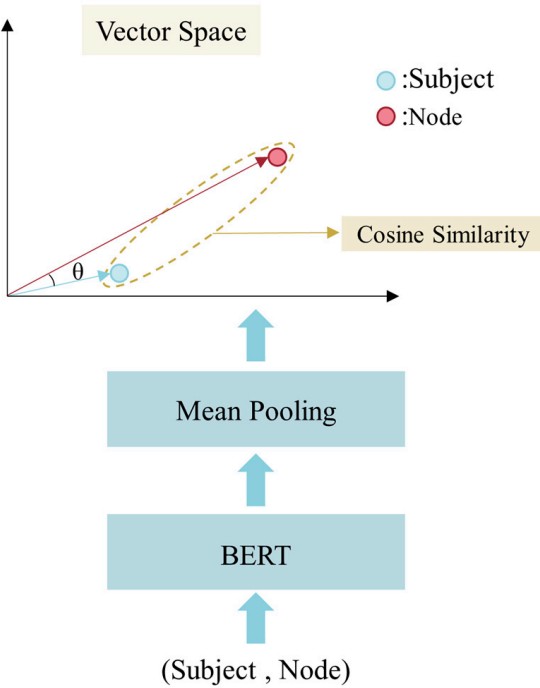

**Figure 9.** Architecture of the SBERT model.

This study prepares 1500 entity alignment training data, which are composed of the common vocabulary pairs in injection molding and annotated with whether they have similar semantics. Entities with similar semantics are labeled as 1 and 0 otherwise. The model's loss value is determined by the cosine similarity predicted by the model and the mean square error (MSE) of the annotations, and AdamW is used as the model optimizer with a 0.001 learning rate to train the model for 200 epochs. Regarding the judgment of whether the subject and node express the same semantics, we use a cosine similarity threshold of 0.7. If the cosine similarity between the node and subject is higher than the threshold, they are considered to express the same semantics. After training, the model achieves an accuracy of 0.947 on the 150 test data. The entity alignment dataset prepared in this study includes many common injection molding terms. Therefore, the performance of the model proves its ability to accurately distinguish most injection molding entity pairs with similar semantics in the test data, and the model's performance will improve with an increase in the amount of training data in the future.

### 3.3. Searching Engine

Our previously proposed search mechanism [35] involved manually setting the search rules to retrieve the information from the knowledge graph. However, this process of setting search rules is time-consuming and can only find the related knowledge of a subject. It is unable to recognize the relationships between this knowledge and the subject. Therefore, in this study, we propose a new knowledge graph retrieval method based on the Transformer model.

After identifying the corresponding node of a subject in the knowledge graph through entity alignment, the system uses a search engine to retrieve the knowledge related to that node based on the graph structure starting from that node. The search results are then filtered based on the semantic of the user's question to find the corresponding knowledge. As shown in Figure 10, we aim for the search engine to retrieve the knowledge related to the starting node and classify it based on the search path.

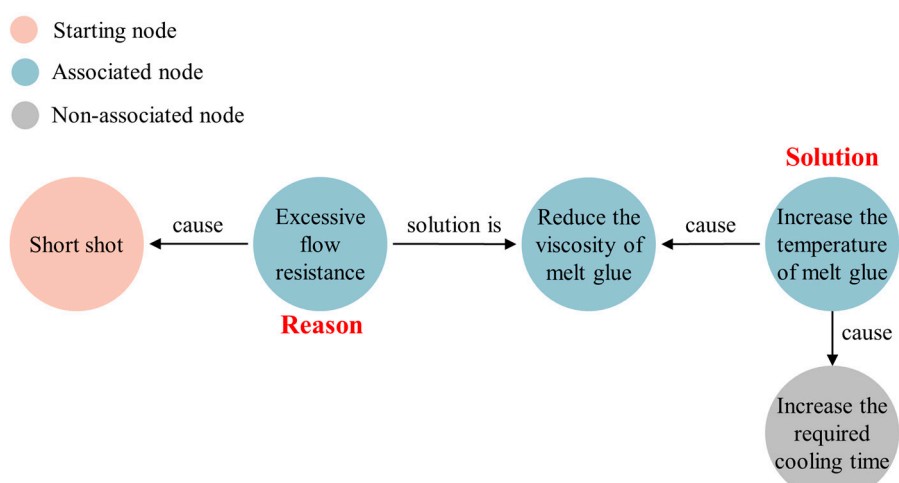

**Figure 10.** Search diagram.

In terms of the training data construction, we encode the paths according to the encoding rule shown in Table 2, save the search paths as sequence data, and the overall process of this is shown in Figure 11.

**Table 2.** Path encoding rule.

| Object | Code |
|:---:|:---:|
| Node | 1 |
| Cause (forward) | 2 |
| May-cause (forward) | 3 |
| Solution is (forward) | 4 |
| Cause (reverse) | 5 |
| May-cause (reverse) | 6 |
| Solution is (reverse) | 7 |

After successfully expressing the search path in the form of sequence data, the model can classify the nodes in the path by reading the sequence data. This study divides the nodes in the path into four categories: impact, cause, solution, and non-associated nodes. We can determine the relationship between a node and its starting node based on the classification results of the model. We will use a 4-dimensional one-hot vector to label the node category in the path as training data, and the labeling method for this is shown in Figure 12.

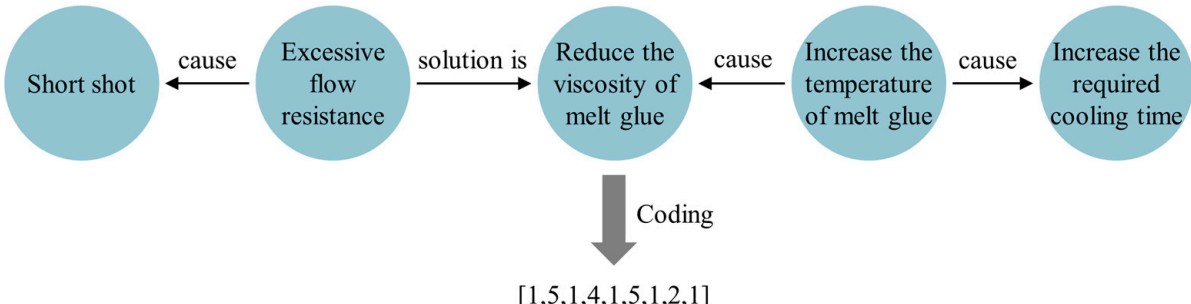

**Figure 11.** Encoding example.

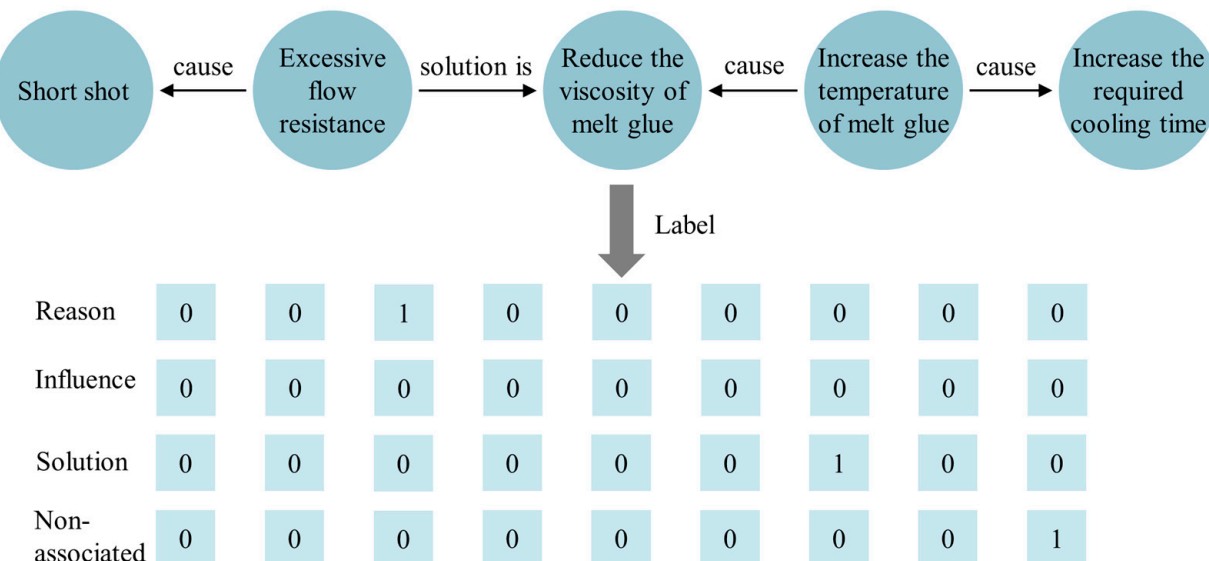

**Figure 12.** Example of sequence data labeling.

This study adopts a 6-layer Transformer Encoder with 4 Attention Heads to interpret the sequence data and generate a 32-dimensional feature array. The model architecture is shown in Figure 13. The feature generated by the 6-layer Transformer Encoder will be fed into a Fully Connected Layer with Sigmoid as its activation function, in order to produce a 4-dimensional one-hot array as the prediction result. The model is trained on 1100 training data for 100 epochs, with a learning rate of 0.0001. After training, the model achieves a prediction accuracy of 1.0 for non-related nodes and a prediction accuracy of 0.98 for the other three categories on 100 validation data.

This study uses the Python programming language to establish a search engine that searches the graph based on loops, as shown in Figure 14. The search engine obtains the neighbor nodes of the starting node through MongoDB and inputs the paths between the starting node and each neighbor node into the model for a prediction through the loop. If the model predicts that the neighbor node is a non-associated node, the node will not be further explored. Otherwise, the neighbor node not in the current path will be obtained, and the above actions will be repeated until all the loops are completed, indicating the completion of the search. The detailed code can be found in Figure 15.

### 3.4. Case Verification

Figure 16 shows the search results for the question: "Knowledge about silver line." The subject recognition model successfully identifies "silver line" as the subject of the question. Additionally, with the help of the entity alignment model, the system is able to identify a node in the knowledge graph named "silver streak", which has a high semantic similarity to the question's subject. The semantic classification model interprets that the

user wants to search for all the relevant knowledge related to "silver line". As a result, the search engine returns all the relevant knowledge about "silver streak" in the form of graphical data, including its cause and solution.

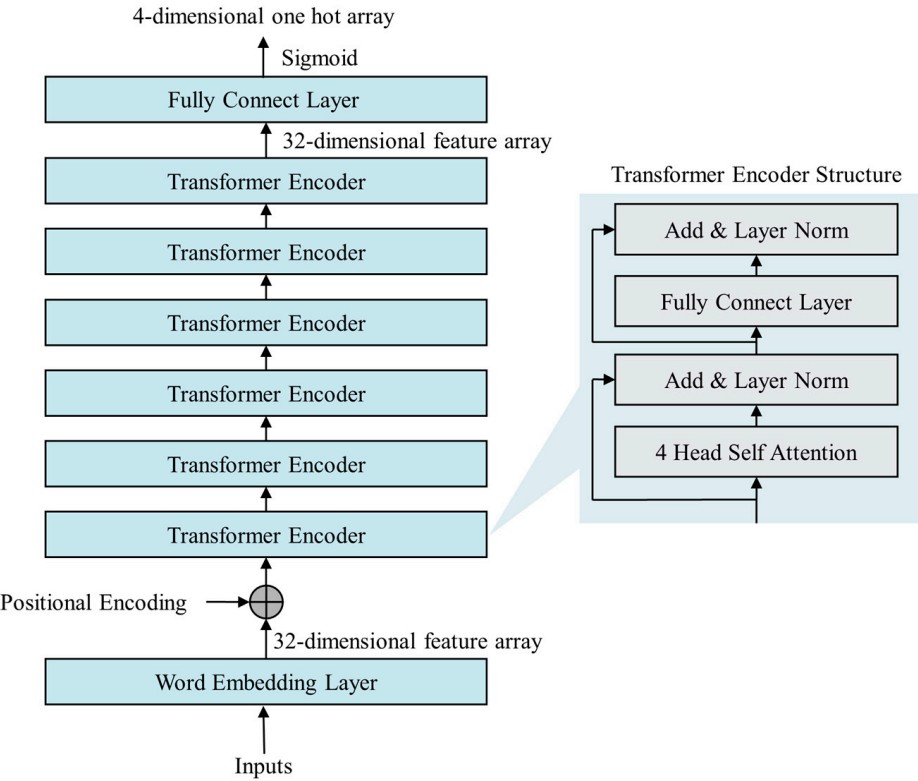

**Figure 13.** Architecture of the searching model.

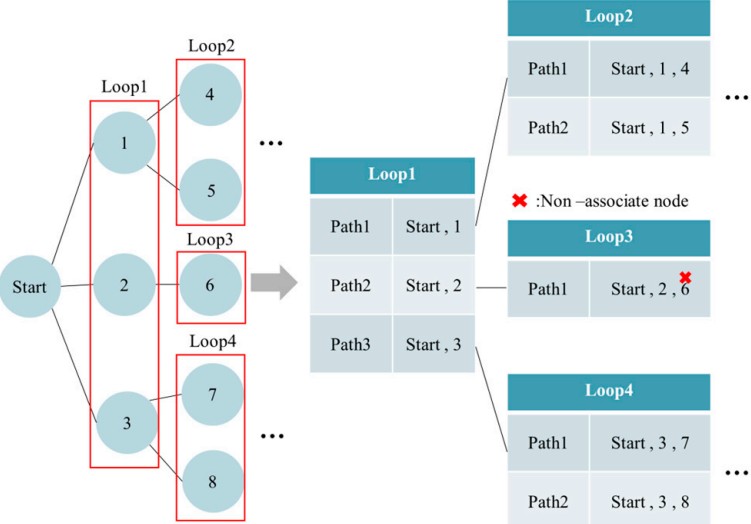

**Figure 14.** Searching engine operation process.

Figure 17 shows the search result for the question: "Reason for silver line". Through the semantic classification model, the system can understand that the user wants to know the reason for "silver line". Therefore, the search engine returns the relevant knowledge about the cause of "silver streak" to the interface.

```python
def search_engine(nodetitle,targetrelation,lang):
    allpath=[]
    nodeid=''

    if lang=='en':
        nodeid=nodecol.find({"entitle":nodetitle})[0]["nodeid"]
    def search(nodeID,lan,path=[],pathlabel=[],graph=[]):
        node=nodecol.find({"nodeid":nodeID})
        nodetype=node[0]['layer']
        nodetitle=''
        if lan=='ch':
            nodetitle=node[0]['title']
        if lan=='en':
            nodetitle=node[0]['entitle']
        if len(path)==0:
            path.append(nodeID)
            pathlabel.append(nodetype)
            graph.append(nodetitle)
        else:
            edge=relationcol.find({"nodeid":path[-1]})
            allrelatednode=edge[0]['totalconnect']
            postedge=edge[0]['postiveconnect']
            for i in allrelatednode:
                if i['node'] == nodeID:
                    relation=i['relation']
                    if nodeID in postedge:
                        relation=relation+'(forward)'
                    else:
                        relation=relation+'(reverse)'
                    graph.append(relation)
                    graph.append(nodetitle)
                    pathlabel.append(relation)
                    pathlabel.append(nodetype)
                    path.append(nodeID)
                    break
        enginepath,classpre,stop=search_pre(pathlabel,model)
        if stop ==1:
            graph=graph[:-2]
            path=path[:-1]
            pathlabel=pathlabel[:-2]
            return path,classpre,pathlabel,graph
        nextstep=relationcol.find({'nodeid':node[0]['nodeid']})[0]['totalconnect']
        postconnect=relationcol.find({'nodeid':node[0]['nodeid']})[0]['postiveconnect']
        reverseconnect=relationcol.find({'nodeid':node[0]['nodeid']})[0]['reverseconnect']
        for i in nextstep:
            if i['node'] not in path:
                path,classpre,pathlabel,graph=search(i['node'],lan,path,pathlabel,graph)
        lista={'path':path,'classpre':classpre,'pathlabel':pathlabel,'graph':graph}
        allpath.append(lista)
        path=path[:-1]
        graph=graph[:-2]
        pathlabel=pathlabel[:-2]
        return path,classpre,pathlabel,graph
```

**Figure 15.** The code for search engine.

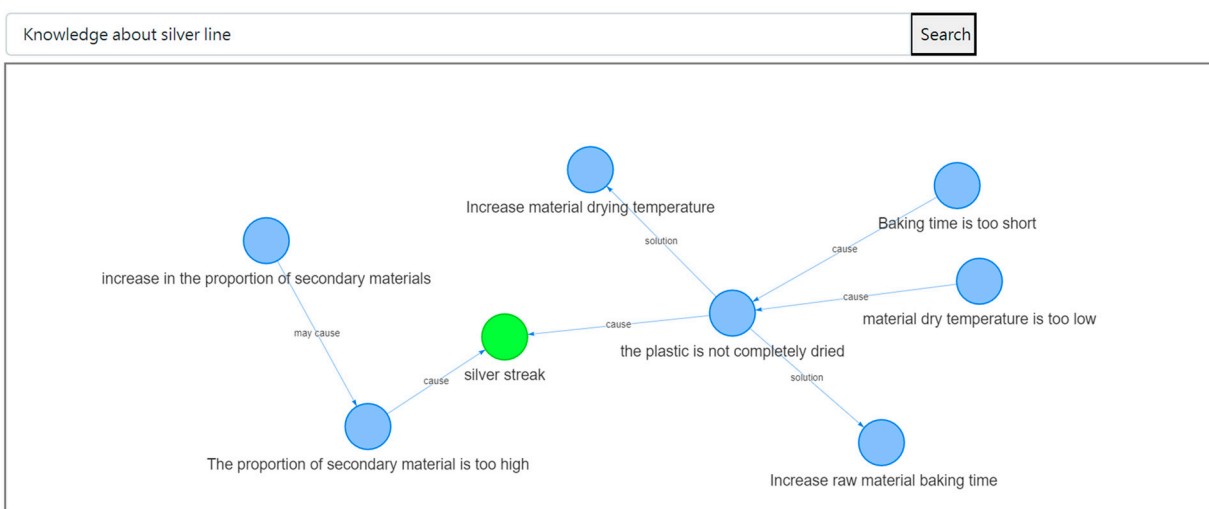

**Figure 16.** Relevant knowledge about silver streak.

Figure 18 shows the search result for the question: "How to solve silver line", and the semantic classification model recognizes that the user is looking for the solution for "silver line". Therefore, the search engine returns the relevant knowledge on how to solve "silver streak" to the interface.

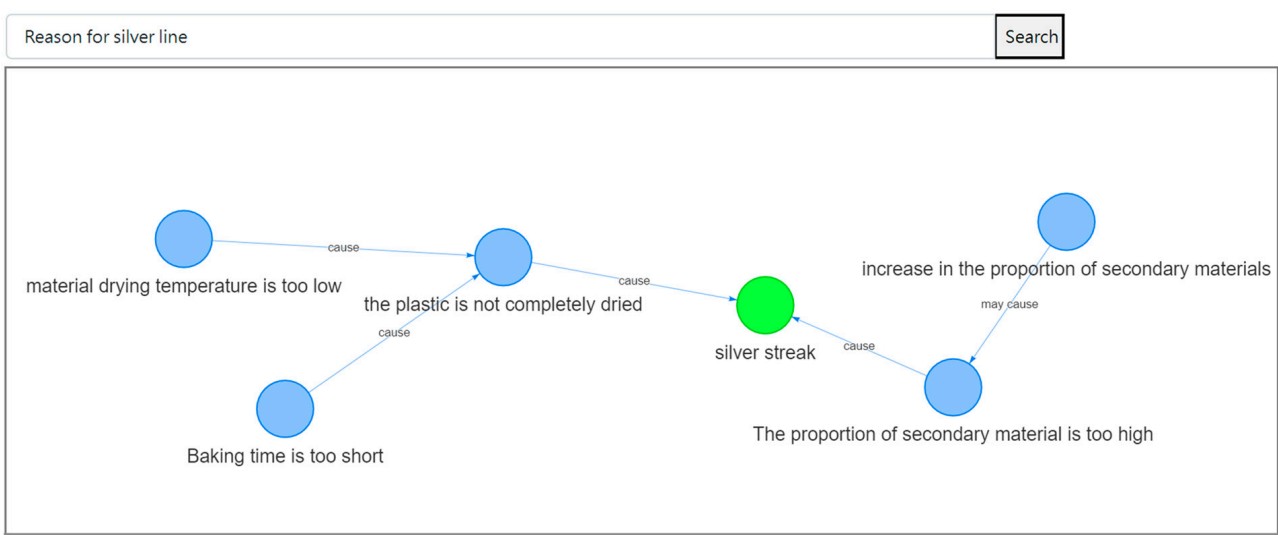

**Figure 17.** Related knowledge about the cause of silver streak.

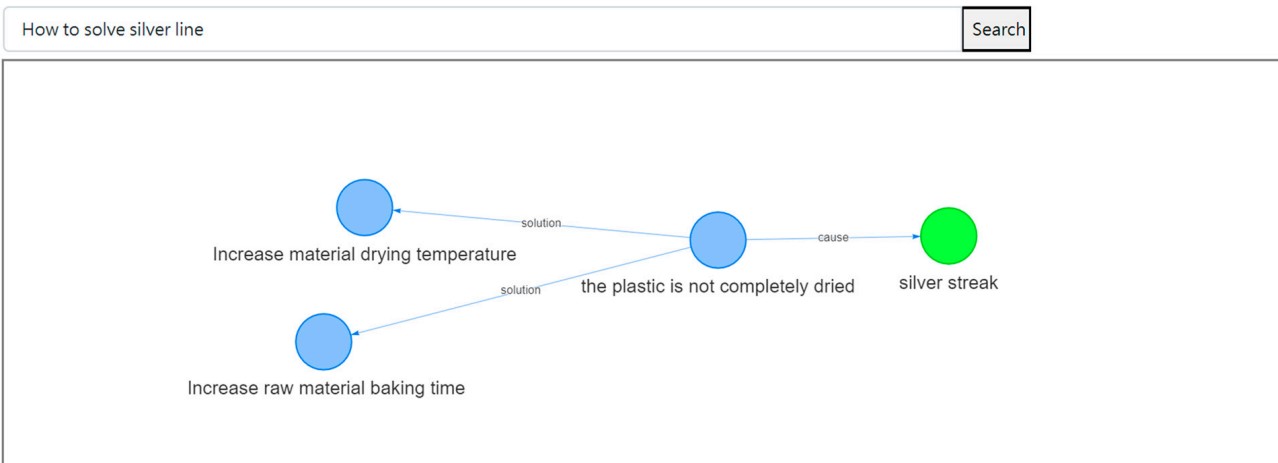

**Figure 18.** Knowledge about how to solve silver streak.

Figure 19 shows the search results for the question: "about short shot", where the system returns all the knowledge related to "short shot" through a semantic classification. The search result contains a total of 65 entities. Figure 20 shows the search results for the user query: "short shot solution", where the system returns the knowledge relevant to solving "short shot". The search result contains 34 entities. By comparing these two search results, it can be seen that our developed system, which understands the user's intent, can filter out unnecessary information and find more relevant knowledge from the related entities of the main subject, unlike the search mechanism proposed in our previous work [35], which can only find all the related knowledge of the subject.

We will evaluate the performance of the search engine by calculating the F1_score for the aforementioned cases. The F1_score is an evaluation metric calculated based on the number of true positives (TP), false positives (FP), and false negatives (FN). Its calculation method is defined in Equation (1). In our case, we consider the number of nodes that are relevant to the user's question and included in the search results as TPs, the number of nodes that are irrelevant to the user's question but included in the search results as FPs, and the number of nodes that should have been included in the search results but were not, as FNs. After computation, the search engine achieved an F1_score of 1.0 for all the search cases mentioned. This indicates that the search engine can effectively identify all the relevant knowledge related to the user's question and filter out unnecessary information. Through these tests, we demonstrate that the system can effectively understand user intent

and retrieve relevant knowledge. It also confirms that conducting deeper searches based on user semantics, rather than retrieving all the knowledge related to a subject, can yield search results that better meet user requirements.

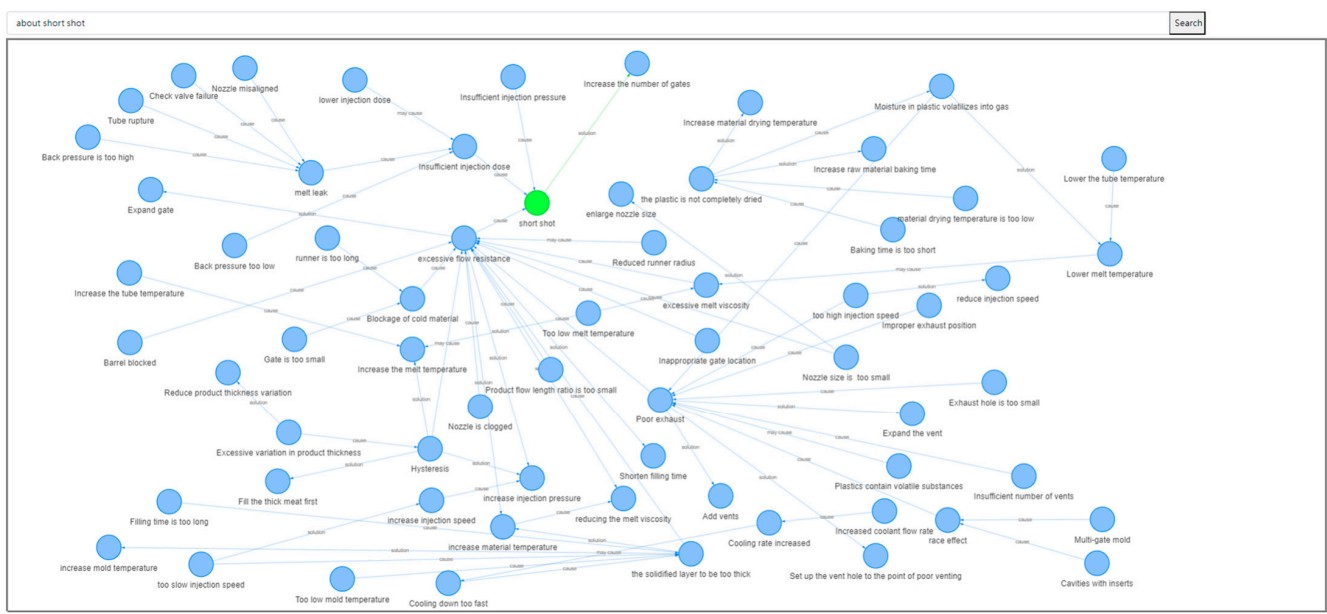

**Figure 19.** Search results for "about short shot".

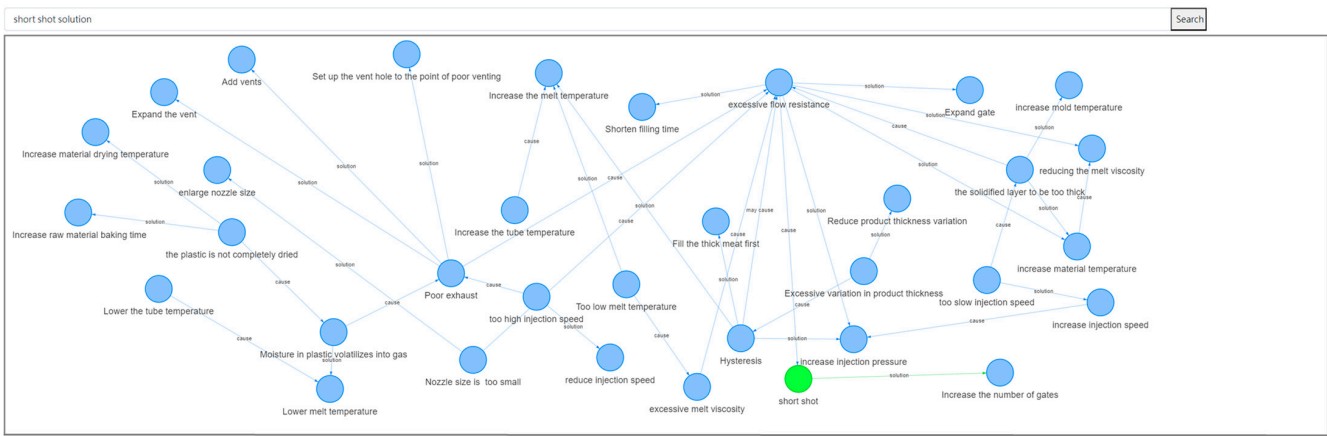

**Figure 20.** Search results for "short shot solution".

$$F1\_Score = TP/(TP + 1/2(FP + FN)) \tag{1}$$

Finally, we also trained the Chinese versions of the three natural language models mentioned above, using Chinese BERT. However, due to the smaller training data, they did not demonstrate the same level of performance as the English version models. However, they still exhibited a certain level of performance. We believe that, with the introduction of different language versions of BERT and sufficient training data, it is possible to train natural language models that exhibit an excellent performance.

## 4. Discussion

This study utilizes natural language technology to understand user questions and develop a search engine that can find relevant knowledge based on user intent. The performance of the subject recognition model, semantic classification model, and entity alignment model on the validation dataset, with accuracies of 0.953, 1.0, and 0.947, respectively,

demonstrates that the system can understand the semantics of most user input questions. The searching model, built using a multi-layer Transformer Encoder, can classify search results based on search paths. In the validation dataset, the model achieved an accuracy of 1.0 in identifying unrelated nodes and 0.98 in knowledge classification. By integrating the above model, this system can help engineers without knowledge graph backgrounds to quickly and easily search a knowledge graph.

The search mechanism we propose will be implemented by building a web server, enabling engineers to access and retrieve information from a knowledge graph through the frontend interface we develop using their computers or mobile devices. However, as the injection molding industrial knowledge graph is still in its development stage, there are currently no plans for its commercialization.

In the future, we hope that the injection molding industry knowledge graph can play a role in eliminating product defects. Therefore, we will develop an automated defect elimination function based on the injection molding industry knowledge graph and integrate it with the platform proposed in this study. The system will be able to automatically identify solutions to product defects based on the process conditions and provide users with a clear understanding of the selection criteria for the solutions. This will be the focus of our future research.

**Author Contributions:** Conceptualization, Z.-W.Z. and W.-R.J.; methodology, Z.-W.Z., W.-R.J. and Y.-H.T.; software, Z.-W.Z. and Y.-H.T.; validation, Z.-W.Z., W.-R.J., Y.-H.T. and M.-C.C.; data curation, Z.-W.Z., W.-R.J. and S.-C.C.; writing—original draft preparation, Z.-W.Z.; writing—review and editing, W.-R.J., S.-C.C. and M.-C.C.; visualization, Z.-W.Z., S.-C.C. and M.-C.C.; supervision, W.-R.J., Y.-H.T., S.-C.C. and M.-C.C. All authors have read and agreed to the published version of the manuscript.

**Funding:** This research received no external funding.

**Data Availability Statement:** Due to the need for future research, we do not have plans to provide the data.

**Conflicts of Interest:** The authors declare no conflict of interest.

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
