# Peer review of "Retrieval of Injection Molding Industrial Knowledge Graph Based on Transformer and BERT"

_applsci, doi:10.3390/app13116687_

Round 1
Reviewer 1 Report
The authors present a method for processing user questions to retrieve relevant knowledge (mainly cause-effect relationships) about the injection molding process using knowledge graphs. The corresponding knowledge graphs have already been presented in [1] and [2]. Overall, the research topic is very interesting as it enables engineers to retrieve information from a knowledge graph based on natural language and without programming skills.
The work is well structured, and the introduction is clear. Some questions and comments:
- The resolution of Figure 1 is insufficient. I would recommend to use a higher resolution figure.
- Figure 9 is similar to Figure 11 in [1].
- In chapter 3.2.2. it is described that the semantic classification model uses the position of the subject to classify the semantic meaning of user's question into the classes "impact", "solution", "reason" or "all". Two questions on that:
· Could it be useful to consider additional information when classifying? Identifying key words such as "how", "reason", "why", "lead to" could potentially help with classification.
· What exactly does the class "all" mean? Can examples be given?
- In chapter 3.2.1. and 3.2.2. it is described that the models for subject recognition and classification achieved an accuracy of 95.3% and 100% respectively. The same database was used for both models. Some questions about the database:
· It is not entirely clear which data are included in the training and validation database. What complexity do the sentences have? Do the sentences exclusively contain one subject or more?
· Is it possible to determine performance indicators regarding the generalizability of the models?
- The chapter “3.2.3 Entity Alignment” shows a high similarity to the chapter “3.2 Entity Alignment” in [2]. Here, the SBERT algorithm is described in both publications and the wording partly coincides. How does the SBERT algorithm differ from the investigations in [2]?
- Figure 10-14 visualize very clearly how the searching engine operation process works. In Figure 14, the Python code is hard to read. I would recommend to use a higher resolution figure.
- Can the developed models be applied to other languages? Are differences in the performance of the models to be expected depending on the language?
- On page 17 it is described that the models were evaluated by a manual verification. I would recommend to use some statistical metrics to get additional information about the method capability.

Author Response
Wen-Ren Jong, Professor
Department of Mechanical Engineering
Chung Yuan Cristian University
Chung-Li, 32023 Taiwan, R.O.C.
Tel: 886(3) 2654353
Fax: 886(3) 2654399
E-mail: august@cycu.edu.tw
2023/05/15
Dear Reviewer,
Thank you for your suggestion. We have improved the resolution of Figure 1 and made modifications to the presentation of Figure 9. In lines 298-302, we introduced the classification criteria for semantic categories. The current semantic classification model determines the semantic meaning of user questions by interpreting the complete question, so the presence of keywords is helpful for the model's determination. Furthermore, we provided a more detailed description of the dataset we prepared in lines 275-278 and 324-326.
The SBERT model we used in this study is the same as the one used in our previous research, as explained in lines 349-356. We also separated the code content from Figure 14 and presented it independently in Figure 15 for better readability. In terms of validation, we will evaluate the performance of the search engine by calculating the F1 score, as discussed in lines 461-474.
Finally, in lines 475-480, we present our perspective on the applicability of our developed model to other languages.
Wen-Ren Jong
Professor
Department of Mechanical Engineering
Chung Yuan Christian University

Reviewer 2 Report
This manuscript tries to figure out a way to solve quality issues during the injection molding process.
I recommend that this paper can be published after revision as indicated.
1) From Figure 15 to Figure 19, the text is too small to see. Due to unclear visibility, it is impossible to determine whether the processing result is correct.
2) As the author suggests, this search engine can help optimize injection molding quality. But can the author explain how engineers can use it? Is it supposed to be sold?
Author Response
Wen-Ren Jong, Professor
Department of Mechanical Engineering
Chung Yuan Cristian University
Chung-Li, 32023 Taiwan, R.O.C.
Tel: 886(3) 2654353
Fax: 886(3) 2654399
E-mail: august@cycu.edu.tw
2023/05/15
Dear Reviewer,
Thank you for your suggestion. We have adjusted the sizes of Figures 16 to 20 to improve readability without affecting the overall layout. However, due to the large volume of search results in Figures 19 and 20, it becomes challenging to assess their accuracy through figure alone. Therefore, in lines 461-474, we explain the method we use to calculate the correctness of search results. Additionally, in lines 492-496, we describe how engineers can use our proposed search engine and share our perspective on commercializing the current research outcomes.
Wen-Ren Jong
Professor
Department of Mechanical Engineering
Chung Yuan Christian University
